# Research on Spatial Correlation Network Structure of Inter-Provincial Electronic Information Manufacturing Industry in China

**Yangjingjing Zhang and Zhi Li \***

Business School, Sichuan University, Chengdu 610065, China; zhangyangjingjing@hotmail.com
**\*** Correspondence: zhil1090@scu.edu.cn

**Abstract:** Electronic information manufacturing industry is the pillar industry of China's economic development. It is of great significance to promote the optimal allocation of national resources and to realize the sustainable development of regional economy by deeply analyzing the spatial correlation structure characteristics of inter-provincial electronic information manufacturing industry, clarifying the status and role of each province in the overall network, and exploring the spillover effect of network space. Based on the panel data of China's inter-provincial electronic information manufacturing industry from 2007 to 2016, this paper constructs a spatial correlation network by using gravity model and deconstructs the spatial network correlation characteristics of inter-provincial electronic information manufacturing industry by using social network analysis. The research shows that the spatial network connectedness of inter-provincial electronic information manufacturing industry is on the rise during the sample period, but the density value is low. The network accessibility and robustness is strong. Compared with the central and western regions, the eastern coastal provinces have stronger control and the ability to attract external resources. Finally, from the perspective of overall situation, differences, and resource support, the paper puts forward some countermeasures and suggestions to promote the sustainable and coordinated development of China's region.

**Keywords:** electronic information manufacturing industry; social network analysis; spatial correlation

## 1. Introduction

With the arrival of the information age and the vigorous development of big data and artificial intelligence, the electronic information industry has become an important engine of the world's economic development and is also a strategic, basic and leading pillar industry of China's national economy [1]. Under the new situation that China accelerates the transformation from a large manufacturing country to a strength manufacturing country, as the material foundation and main part of the electronic information industry, electronic information manufacturing industry plays a supporting role in intelligent manufacturing and industrial transformation and upgrading. It is the key force to realize the goal of "Made in China 2025" and the "13th Five-Year Plan for the Development of National Strategic Emerging Industries" [2]. Among them, "Made in China 2025" is a measure issued by the State Council of China in 2015 to comprehensively upgrade China's industry. This course of action was inspired by Germany's "Industry 4.0" plan [3]. However, compared with Germany, China is still in the process of industrialization, with relatively weak technology and incomplete industrialization. Compared with advanced countries, China still has a big gap [4]. Therefore, in order to achieve the goal of becoming a world leading industrial power by 2049 [5], it is particularly important to strengthen the research on electronic information manufacturing industry.

However, from the current development point of view, due to the differences in economic development level, resource endowment and geographical location of various provinces, the regional

development of China's electronic information manufacturing industry is quite different. In 2016, the main business income of the electronic information manufacturing industry in the eastern region accounted for 74.7% of the national total, and the scale of research and development personnel accounted for 70.5% of the national total. On the one hand, unbalanced regional development and widening regional disparities are not conducive to the rational allocation of resources, the rational distribution of industries and the complementary advantages of various regions. It is not conducive to the long-term sustainable and stable development of the national economy. The goal of sustainable and coordinated development of regional economy is to gradually narrow the regional disparities on the basis of steadily improving the overall benefits of the national economy and finally achieve balanced development [6]. On the other hand, unbalanced regional development does not mean that regions are divided and do not affect each other [7]. In fact, the spatial correlation of the development of China's inter-provincial electronic information manufacturing industry has shown a systematic and complex network structure. From the perspective of government policies, the country attaches great importance to regional coordinated development and has formulated corresponding strategies for regional coordinated development of electronic information manufacturing industry, so as to give full play to regional advantages and promote the correlation between the development of electronic information manufacturing industry in various provinces. From the perspective of market mechanism, the capital, technology, talents and other factors of production in various provinces have realized cross-regional resource flow, which has greatly enhanced the spatial correlation of the development of inter-provincial electronic information manufacturing industry. The purpose of reasonable resource allocation is to achieve social equity, stable ecological environment, balanced social economy and sustainable development [8]. Therefore, studying the spatial correlation structure of inter-provincial electronic information manufacturing industry can not only master the evolution trend of the overall network spatial pattern, but also clarify the position and role of each province in the spatial correlation network of electronic information manufacturing industry and the spillover effect of network space inside and outside the region. Thus, it is possible to implement regional industrial policies in a targeted way and promote inter-provincial industrial agglomeration and technology spillover of electronic information manufacturing industry. It is conducive to improving the optimal allocation level of national resources and finally realizing the sustainable development of regional economy.

So far, scholars have carried out extensive and in-depth exploration on the electronic information manufacturing industry from various aspects. For example, Yang Huanjin et al. have revealed the existing problems in human resources management, environmental resources and industrial structure in the development of China's electronic information industry from the perspective of current situation analysis and put forward corresponding countermeasures [9]. Gou Zhongwen discussed the operation mechanism of the innovation mechanism of China's electronic information industry from the perspective of industrial competitiveness and put forward the idea of improving the innovation system [10]. Tao Feng et al. analyzed the upgrading mode of OEM (Original Equipment Manufacture) in the electronic information manufacturing industry from the perspective of value chain in order to explain the transformation and upgrading of China's OEM industry [11]. Weiliang Chen et al. constructed a fusion evaluation model of China's electronic information industry and financial industry from the perspective of industrial integration, and empirically measured the impact of industrial integration level on the competitiveness of the electronic information industry [12].

Although academic circles have done a lot of research on electronic information manufacturing industry, the analysis of its spatial correlation is relatively scarce. Relevant documents mainly focus on regional agglomeration and spatial evolution. From the general law of industrial development, manufacturing industry has obvious spatial agglomeration characteristics, while electronic information manufacturing industry, as a capital-intensive and technology-intensive industry, is more typical in agglomeration development [13]. At present, China has formed a relatively perfect industrial chain agglomeration in the Bohai Sea region, the Yangtze River Delta region and the Pearl River Delta region, but the association and cooperation among industrial clusters are still insufficient [14]. Lei Ping studied

the regional agglomeration effect of electronic information manufacturing industry and found that China's electronic information manufacturing industry has not yet formed the cooperative advantages of clusters. The industrial value chain lacks local embeddedness [15,16]. In the research of spatial evolution, MAO et al. empirically analyzes the core-periphery gradient of manufacturing industries across Chinese provinces and assesses the extent to which these provinces have changed in recent years. The results indicated that industrial development differentials across regions arise because of not only the uneven distribution of industries but also the inconsistent evolving trends of industrial structure for each province [17]. Zhu Huasheng analyzed the change trend of industrial ties in content and space after 1970s and the main factors that affect the strength of local ties in industrial clusters in the new era [18]. Hu Wei et al. used the spatial analysis method of geographic information system to analyze the regional development characteristics of China's electronic information manufacturing industry and its pattern evolution within the geographic space since the reform and opening up. The research found that the development trend of the electronic information manufacturing industry in the south was obviously better than that in the north, and showed a gradual migration trend to the southwest [19].

The above research reveals the significant spatial correlation and spatial agglomeration characteristics of China's inter-provincial electronic information manufacturing industry. However, most of the existing research is based on "attribute data" rather than "relational data" [20], which makes the research on spatial correlation network consider the "neighboring" or "proximity" effect in geography, but the conclusions drawn from this are often limited to the geographical neighboring regions. Thus, it is difficult to master the spatial correlation characteristics of inter-provincial electronic information manufacturing industry as a whole. In addition, because relational data do not meet the "independence assumption of variables" in the conventional statistical sense, various multivariate statistical methods in the general sense cannot be used to analyze relational data [21]. But social network analysis can just make up for the above deficiencies. It emphasizes the importance of relational data among nodes in the network, and comprehensively considers the influence of direct and indirect relationships among nodes. It applies a series of algorithms to quantify and describe the aggregation form of nodes in the network, so it can reveal the correlation and structure of the interregional network more intuitively.

As a new research paradigm [22], social network analysis has been widely used in sociology [23,24], economics [25,26], management [27,28]. In addition, some studies use this method to explore regional spatial network research [29,30]. Based on relational data and network perspective, this paper constructs a spatial correlation network model using inter-provincial electronic information manufacturing data from 2007 to 2016, and explores the spatial correlation of China's electronic information manufacturing industry using social network analysis. By measuring network density, network connectedness, network hierarchy and network efficiency, the overall network characteristics and evolution trend are analyzed [31]. Through measuring degree centrality and betweenness centrality, this paper analyzes the position and role of individual provinces in the network and masters the internal connections and differences between regions [32]. Through spatial clustering, the provinces are divided into four blocks, the attribute characteristics of each block and the transmission mechanism of spillover effects within and between blocks are analyzed. Then policy suggestions are put forward to promote the coordinated development and overall promotion of regional industries, so as to provide theoretical guidance and decision-making reference for China's regional sustainable development.

## 2. Methodology and Data

### 2.1. Construction of Spatial Correlation Network in Electronic Information Manufacturing Industry

The "nodes" in the spatial correlation network of inter-provincial electronic information manufacturing industry are provinces. The "lines" in the network are the spatial correlation relationships among provinces, and the determination of relationships is the key to network analysis. According to the existing literature, the construction of spatial correlation network mainly adopts the VAR (Vector Autoregression) Granger Causality test method [33,34] and gravity model [35,36].

However, the network constructed by VAR model cannot describe the evolution trend of spatial correlation network, and is too sensitive to the selection of lag order, which reduces the accuracy of network structure characterization to a certain extent [37]. The gravity model is not only more suitable for gross data, but also can comprehensively consider geographical distance, economic development, population size and other factors to describe the strength of the connection between regions, and then can use cross-section data to describe the evolution trend of the spatial correlation network. Therefore, this paper uses the gravity model of quantitative analysis of regional economic ties to build the spatial correlation network of electronic information manufacturing industry. The modified gravity model is as follows:

$$y_{ij} = k_{ij} \times \frac{\sqrt{R_i E_i} \sqrt{R_j E_j}}{D_{ij}^2} \ , \ k_{ij} = \frac{I_i}{I_i + I_j}. \tag{1}$$

In the formula, $i$ and $j$ represent different provinces; $y_{ij}$ indicates the spatial interaction force between province $i$ and province $j$; $k_{ij}$ indicates the contribution rate of province $i$ in the correlation degree of electronic information manufacturing between province $i$ and province $j$; $R_i$ and $R_j$ respectively represent the main business income of electronic information manufacturing industry above designated size of province $i$ and province $j$; $E_i$ and $E_j$ respectively represent the number of electronic information manufacturing enterprises above designated size of province $i$ and province $j$. In order to consider the influence of geographical distance between provinces on industrial spatial correlation, this paper uses $D_{ij}$ to represent the spherical distance between $i$ and $j$ provincial capitals.

After the gravity matrix of electronic information manufacturing industry between provinces is obtained according to Formula (1), the network relation matrix $X = (x_{ij})_{N \times N}$ can be obtained according to Formula (2). Where $N$ is the number of provinces studied. In view of the availability of data, this paper will study 30 provinces and cities in the country (excluding Tibet, Hong Kong, Macao and Taiwan regions) and construct a relation matrix $X = (x_{ij})_{30 \times 30}$. Since the calculated gravity matrix is asymmetric, the network is a directed spatial correlation network.

$$x_{ij} = \begin{cases} 1, & y_{ij} \geq \frac{\sum_{j=1}^{N} y_{ij}}{N} \\ 0, & y_{ij} < \frac{\sum_{j=1}^{N} y_{ij}}{N} \end{cases} \tag{2}$$

*2.2. Characterization of Spatial Correlation Networks in Electronic Information Manufacturing Industry*

The Social network analysis method [38] is used to study the structural characteristics of inter-provincial electronic information manufacturing spatial correlation network, including overall network characteristics, network characteristics of each node, and block model analysis.

2.2.1. Overall Network Characteristics

(1) Network density. It refers to the ratio of the actual number of relationships between provinces in the network to the maximum possible number of relationships across the network, reflecting the degree of density of network relationships. The more relationships there are, the higher the network density will be, and the closer the spatial correlation between provinces will be. The formula is: $D = M/(N(N-1))$, where $M$ represents the actual number of relationships contained in the network, and $N$ represents the maximum possible value of the total number of relationships contained in theory.

(2) Network connectedness. It reflects the robustness and vulnerability of the overall network. If any pair of provinces can reach each other in the network, the connectedness is 1, which indicates that the network has strong correlation and robustness. If all the connections are linked through a certain province, the network is highly dependent on this province. Once this province is excluded, the network may collapse, thus it is not robust. The formula is: $C = 1 - \left[ \frac{V}{N(N-1)/2} \right]$, where $V$ represents the number of unreachable node pairs in the network, and $N$ represents the size of the network.

(3) Network hierarchy. It describes the extent to which network nodes are asymmetrically reachable, reflecting the dominant position of provinces and cities. The higher the hierarchy, the more obvious the "core-edge structure" of the network will be. The formula is $GH = 1 - V/max(V)$, where $V$ represents the number of symmetrically reachable pairs in the network, and $max(V)$ represents the number of node pairs that $i$ can reach $j$ or $j$ can reach $i$.

(4) Network efficiency. It refers to the extent to which there are redundant lines in the graph on the premise of ensuring accessibility, reflecting the connection efficiency of various provinces in the network. The lower the network efficiency value is, the more redundant lines there are, and the more robust the network will be. The formula is $GE = 1 - V/max(V)$, where $V$ represents the number of redundant lines and $max(V)$ represents the maximum possible number of excess links.

### 2.2.2. Network Characteristics of Each Node

(1) Degree centrality. It reflects the central position of a single province in the spatial correlation network. The higher the degree of centrality, the more connections the provinces have with other provinces. The higher the status in the network, the greater the influence and control on other provinces. The formula is:

$$C_{i(in)} = \sum_{j=1}^{n} R_{ij(in)}, \tag{3}$$

$$C_{i(out)} = \sum_{j=1}^{n} R_{ij(out)}. \tag{4}$$

In the formula, $C_{i(in)}$ and $C_{i(out)}$ represent the in-degree centrality and out-degree centrality of unit $i$ respectively, $R_{ij}$ is the spatial interaction force of spatial unit $i$ and spatial unit $j$.

(2) Betweenness centrality. It measures the extent to which spatial unit control resources. If a node is on the shortest path of many other node pairs, the node has a higher betweenness centrality. The formula is:

$$B_i = \sum_{i=j\neq k} \frac{D_{jk}(i)}{D_{jk}}. \tag{5}$$

In the formula, $B_i$ represents the betweenness centrality of unit $i$, $D_{jk}$ is the number of shortcuts existing in unit $j$ and unit $k$, $D_{jk}(i)$ is the number of shortcuts existing in unit $j$ and unit $k$ that pass through unit $i$.

### 2.2.3. Spatial Clustering—Block Model

Block model analysis is mainly used to describe the roles and positions of provinces in spatial correlation networks. Block model was first proposed by White et al. [39]. It is a research method to analyze the location characteristics of network node groups. According to Wasserman and Faust's measurement method of evaluation network block model [40], this paper divides the spatial correlation network of electronic information manufacturing industry into four blocks: "net beneficial block", "net spillover block", "broker block" and "bidirectional spillover block" [41]. The meaning of the "net beneficial block" is that there are more internal relations than external relations in this block, and there is little spillover effect on other blocks. The meaning of the "net spillover block" is that the members of this block send more relations to other blocks and accept less external relations. The meaning of the "broker block" is that the block acts as an "intermediary" in the network. It can accept external relations and send relations to the outside world, but its internal member relations are few. "Bidirectional spillover block" means that the block sends out more relationships to other blocks, while members in the block have more mutual relationships but receive less external relationships.

### 2.3. Data

Based on the availability of data, this paper takes 30 provinces and cities in China (excluding Tibet and Hong Kong, Macao and Taiwan regions) as research objects, and empirically examines the spatial correlation of inter-provincial electronic information manufacturing industry. The time span of data is 2007–2016.The total industrial output value of all provinces and cities comes from the China Statistical Yearbook 2007–2016, the main business income and number of enterprises of electronic information manufacturing industry above designated size in 2007–2009 come from the China Electronic Information Industry Statistical Yearbook 1978–2009, the main business income and number of enterprises of electronic information manufacturing industry above designated size in 2009–2016 come from the China Electronic Information Industry Statistical Yearbook 2009–2016, and the geographical distance between provinces is expressed by spherical distance between provincial capitals and measured by ARCGIS10.2 software (Esri, Redlands, CA, USA).

## 3. Results and Discussion

### 3.1. Evolution Trend of Overall Network of Electronic Information Manufacturing Industry in China

According to the modified gravity model, this paper determines the spatial correlation of inter-provincial electronic information manufacturing industry and establishes the relationship matrix. The relation matrix 0–1 is binarized and imported into UCINET software to draw the visual network structure extension figures for 2007 (Figure 1), 2011 (Figure 2) and 2016 (Figure 3). The three figures all present a typical network structure. At the same time, we can intuitively see the process and evolution trend of China's electronic information manufacturing industry from the east to the central and western regions in the past ten years, especially the relative decline trend in Beijing and Shanghai. However, the electronic information manufacturing industry in Jiangsu and Guangdong developed rapidly during the sample period, and the migration trend of the center of gravity during this period was not obvious due to the joint efforts of many parties. In addition, Hainan and Hebei are located at the edge of the network and have less direct contact with other provinces.

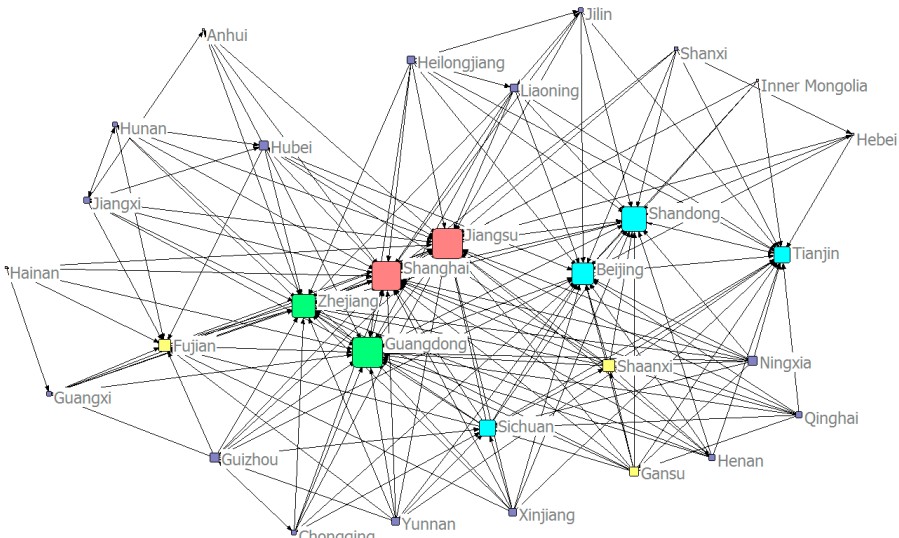

**Figure 1.** Spatial correlation network diagram of inter-provincial electronic information manufacturing industry in 2007.

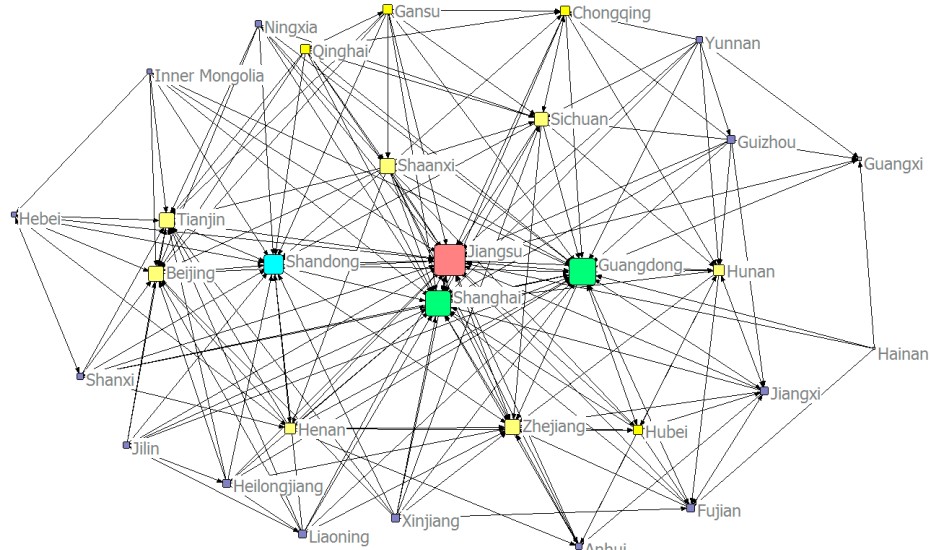

**Figure 2.** Spatial correlation network diagram of inter-provincial electronic information manufacturing industry in 2011.

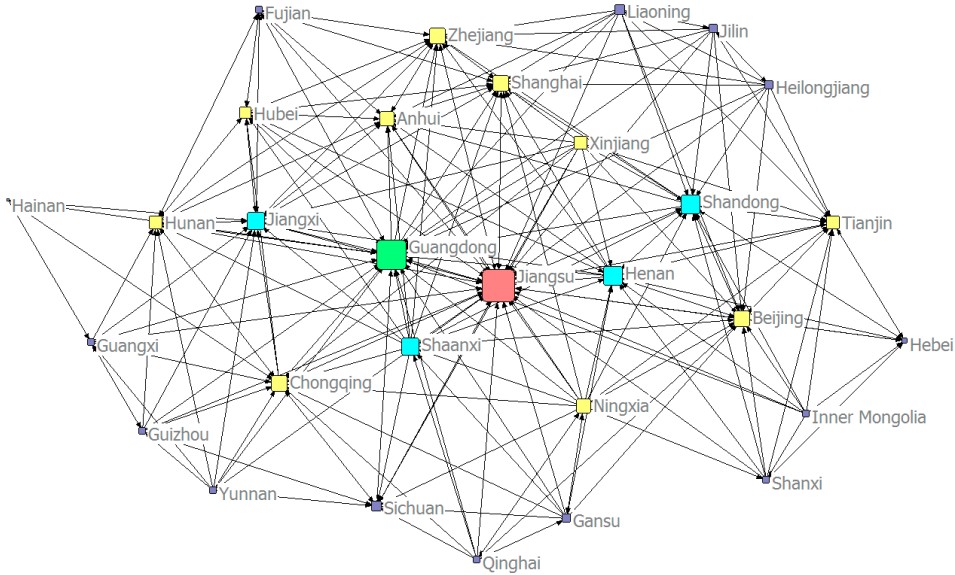

**Figure 3.** Spatial correlation network diagram of inter-provincial electronic information manufacturing industry in 2016.

*3.2. Analysis of the Overall Network Structure Characteristics of China's Electronic Information Manufacturing Industry*

3.2.1. Network Density

As shown in Figure 4, from the overall trend, inter-provincial electronic information manufacturing industry has become more closely linked in space in the past decade, with the number of network relationships rising from 195 in 2007 to 225 in 2016, and the corresponding density rising from 0.22 to 0.26. From the perspective of regional development, with the deepening of economic globalization, factors of production flow freely among different industries and regions. Technological progress and the development of transportation and communication technologies have weakened the role of geographical location in industrial competition and improved the spatial correlation of industrial development. However, at the same time, this paper also found that the relationships and density of

China's inter-provincial electronic information manufacturing network showed slight fluctuations from 2008 to 2013. The reason may be due to the global financial crisis. In 2009, China's electronic information manufacturing industry experienced negative growth for the first time in the new century and became the most impacted industry in the national economy. In 2014, with the gradual recovery of market economic situation in developed countries and the rapid growth of emerging economies, China's electronic information manufacturing industry recovered and its growth rate gradually stabilized.

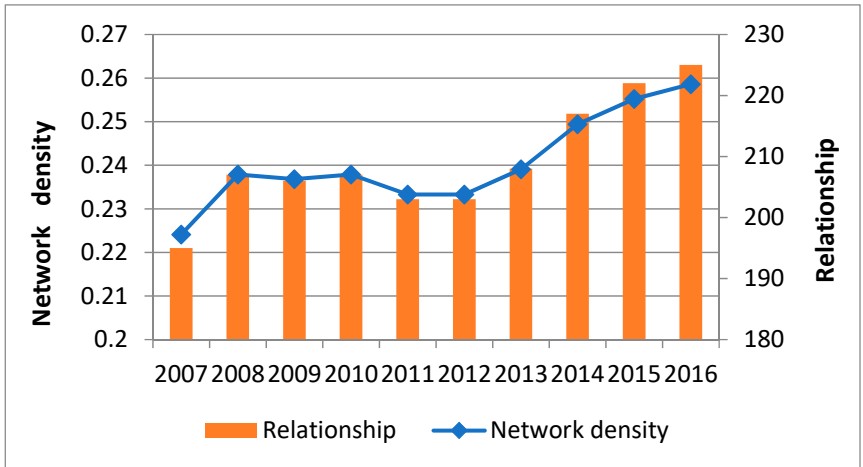

**Figure 4.** Number of spatial relationships and network density of China's electronic information manufacturing industry from 2007 to 2016.

Although the degree of correlation network is on the rise during the sample period, the maximum number of correlation relationships for this network is 225, which is far from the maximum possible number of correlation relationships (900) in 30 provinces. It shows that the inter-provincial electronic information manufacturing industry in China is not highly correlated, and there is still a lot of room for mutual cooperation and exchange. At the same time, too high network density will also increase redundant connections, resulting in low efficiency of the network, resulting in diseconomies of scale, which is not conducive to industrial growth. Therefore, it is necessary to control the network density within a reasonable limit to ensure the spatial optimal allocation of the industry.

3.2.2. Spatial Correlation

In this paper, three indicators of network connectedness, hierarchy and efficiency are used to describe the correlation of inter-provincial electronic information manufacturing spatial correlation network. The measurement results show that the network correlation degree from 2007 to 2016 is 1, which indicates that any pair of nodes can reach each other, and all provinces have direct or indirect spatial spillover effects and obvious spatial correlation. As shown in Figure 5, during the sample period, the hierarchy of the spatial correlation network of China's electronic information manufacturing industry generally showed a downward trend, from 0.751 in 2007 to 0.545 in 2016, which indicates that the differences in the development of the electronic information manufacturing industry between provinces are narrowing and the mutual influence between regions is gradually increasing. This is directly related to the regional openness of the electronic information manufacturing industry in recent years and the government's policy support to regions with weak development level. Except for a few years, the efficiency value of the spatial correlation network for the development of inter-provincial electronic information manufacturing industry from 2007 to 2016 shows an obvious downward trend, from 0.643 in 2007 to 0.618 in 2016. This indicates that inter-provincial redundant connections are increasing and the spatial network is becoming more stable. The spatial overflow channels between nodes are increasing, thus the barriers between regions are gradually broken down. The cooperation

and promotion of electronic information manufacturing industry between provinces can be realized more conveniently through the whole network.

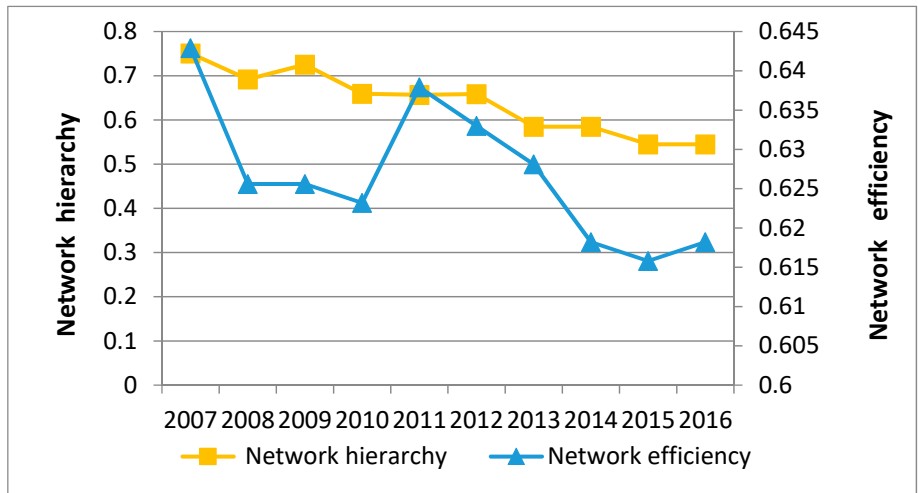

**Figure 5.** Spatial network hierarchy and network efficiency of China's electronic information manufacturing industry from 2007 to 2016.

### 3.3. Analysis of Network Individual Characteristics of China's Electronic Information Manufacturing Industry

In order to analyze the power and status of individual provinces in the spatial correlation network of electronic information manufacturing industry more clearly, this paper makes a network centrality analysis on the degree centrality and betweenness centrality of 30 provinces [42].

#### 3.3.1. Degree Centrality

As shown in the measurement results of degree centrality in Table 1, the average degree centrality of 30 provinces is 12.267, and the centrality of Jiangsu, Guangdong, Shandong, Henan, Shaanxi and other 12 provinces is higher than the mean value, indicating that these provinces have more direct correlation with other provinces in the spatial correlation network of the inter-provincial electronic information manufacturing industry. From a numerical point of view, Jiangsu's degree centrality ranks the highest in the country at 28, indicating Jiangsu's core position and influence in the nation's electronic information manufacturing network. However, Hainan, Hebei and Guangxi ranked lower, which indicates that these provinces have little correlation with other provinces. It may be due to they are in a relatively remote geographical location or have a relatively limited economic scale, resulting in a weak correlation with other provinces.

**Table 1.** Centrality analysis of inter-provincial electronic information manufacturing industry spatial correlation network.

| Region | Degree Centrality | | Betweenness Centrality | |
|---|---|---|---|---|
| | Centrality | Rank | Centrality | Rank |
| Beijing | 15 | 7 | 6.483 | 12 |
| Tianjin | 12 | 13 | 2.7 | 15 |
| Hebei | 7 | 29 | 25.286 | 7 |
| Shanxi | 8 | 22 | 0.2 | 22 |
| Inner Mongolia | 8 | 22 | 0 | 24 |
| Liaoning | 10 | 17 | 0.333 | 20 |
| Jilin | 9 | 19 | 1 | 16 |

**Table 1.** *Cont.*

| Region | Degree Centrality | | Betweenness Centrality | |
|---|---|---|---|---|
| | Centrality | Rank | Centrality | Rank |
| Heilongjiang | 9 | 19 | 0 | 24 |
| Shanghai | 15 | 7 | 0.476 | 18 |
| Jiangsu | 28 | 1 | 100.512 | 2 |
| Zhejiang | 15 | 7 | 0.476 | 18 |
| Anhui | 14 | 11 | 8.84 | 11 |
| Fujian | 8 | 22 | 0.143 | 23 |
| Jiangxi | 16 | 5 | 23.163 | 8 |
| Shandong | 17 | 3 | 70.402 | 3 |
| Henan | 17 | 3 | 35.629 | 5 |
| Hubei | 11 | 16 | 29.082 | 6 |
| Hunan | 12 | 13 | 10.3 | 9 |
| Guangdong | 25 | 2 | 118.983 | 1 |
| Guangxi | 8 | 22 | 10.196 | 10 |
| Hainan | 4 | 30 | 0 | 24 |
| Chongqing | 15 | 7 | 46.135 | 4 |
| Sichuan | 10 | 17 | 0 | 24 |
| Guizhou | 8 | 22 | 1 | 16 |
| Yunnan | 8 | 22 | 0 | 24 |
| Shaanxi | 16 | 5 | 4.81 | 13 |
| Gansu | 9 | 19 | 0.333 | 20 |
| Qinghai | 8 | 22 | 0 | 24 |
| Ningxia | 14 | 11 | 4.517 | 14 |
| Xinjiang | 12 | 13 | 0 | 24 |

### 3.3.2. Betweenness Centrality

According to the measurement results of the betweenness centrality in Table 1, Guangdong has the highest betweenness centrality, which is 118.983. The following are Jiangsu, Shandong, Chongqing, Henan, and Hubei, which reflect that these six regions belong to the core circle in the national spatial correlation network of electronic information manufacturing industry, are more closely connected with other regions. They have greater control over other regions and enjoy more advantages of "intermediary" and "bridge" in the spatial network structure. They have more opportunities to access information and resources and play the role of transmission and center in the spatial correlation network of electronic information manufacturing industry. However, Xinjiang, Heilongjiang, Qinghai and other provinces with a betweenness centrality of 0 are in the peripheral position of the network structure, and their ability to control and influence other units needs to be strengthened. In addition, the betweenness degree of each province in the spatial correlation network of electronic information manufacturing industry is uneven, which shows a remarkable unbalanced form. Most of the connections are realized through major manufacturing provinces such as Guangdong, Jiangsu and Shandong.

### 3.4. Block Model Analysis of Spatial Correlation Network of China's Electronic Information Manufacturing Industry

Through block model analysis, this paper can get the spatial clustering characteristics and action rules of China's inter-provincial electronic information manufacturing network. CONCOR algorithm is adopted to divide 30 provinces into four blocks with a maximum separation degree of 2 and a convergence standard of 0.2. Table 2 can reflect the positions of the four blocks in the spatial correlation network of electronic information manufacturing industry and the attributes of each block. Overall, there are 225 correlations in China's electronic information manufacturing network in 2016, with 106 correlations within the blocks and 119 correlations outside the blocks, indicating that there are significant spatial correlations and spillover relationships between the blocks.

**Table 2.** Spillover effect of spatial correlation block in inter-provincial electronic information manufacturing industry.

| Block | Number of Receiving Relations | | Number of Sending out Relations | | Expected Internal Relationship Ratio (%) | Actual Internal Relationship Ratio (%) | Block Properties |
|---|---|---|---|---|---|---|---|
| | Intra Block | Out of Block | Intra Block | Out of Block | | | |
| Block 1 | 30 | 26 | 30 | 18 | 20.7 | 62.5 | Bidirectional spillover block |
| Block 2 | 14 | 0 | 14 | 69 | 24.1 | 16.9 | Net spillover block |
| Block 3 | 43 | 76 | 43 | 8 | 24.1 | 84.3 | Net beneficial block |
| Block 4 | 19 | 17 | 19 | 24 | 20.7 | 44.2 | Broker block |

Among them, there are 7provinces located in block 1, namely Beijing, Tianjin, Hebei, Shanxi, Inner Mongolia, Henan and Shandong, which are mainly distributed in north China. There are 48 total spillovers in block 1 and 30 intra-block spillovers. The expected ratio of internal relation is 20.7%, while the actual ratio of internal relation is 62.5%, which belongs to the "bidirectional spillover block". This block has a relatively large number of connections from members inside the block, and at the same time sends out a large number of relationships to both the inside and the outside. The bidirectional spillover effect of block 1 is obvious.

There are 8provinces in block 2, namely Jilin, Liaoning, Qinghai, Shaanxi, Gansu, Xinjiang, Ningxia and Heilongjiang, which are mainly distributed in northeast and northwest regions. There are 83 spillovers of block 2 and 14 relationships within the block. Block 2 does not receive spillovers from other blocks. The expected ratio of internal relation is 24.1%, while the actual ratio of internal relation is 16.9%, which belongs to the "net spillover block". The number of spillovers between this block and other blocks is significantly higher than the number of spillovers it receives from external blocks. It indicates that the local electronic information manufacturing industry develops slowly, which causes the external flow of resources, and the intercommunication and cooperation of electronic information manufacturing industry within the block still need to be strengthened.

There are 8 provinces in block 3, namely Jiangsu, Hubei, Fujian, Guangdong, Anhui, Jiangxi, Shanghai and Zhejiang, which are mainly distributed in eastern China. The spillover relation of block 3 to the outside is 8 and block 3 received 76 spillovers from other blocks. The expected ratio of internal relation is 24.1%, while the actual ratio internal relation is 84.3%, which belongs to the "net beneficial block". This block mainly accepts spillovers from other provinces, and its relationship number is significantly higher than that sent to external blocks. For example, Jiangsu, Guangdong, Shanghai and Zhejiang are all coastal provinces with developed economy. With rich scientific and technological development resources, they will benefit more from industrial development.

There are 7 provinces in block 4, namely Hunan, Guizhou, Yunnan, Guangxi, Hainan, Chongqing and Sichuan, which are mainly distributed in the southwest. Block 4 has 24 spillovers to the outside and receives 17 spillovers from other blocks. The expected ratio of internal relation is 20.7%, while the actual ratio internal relation is 44.2%, which belongs to the "broker block". This block not only accepts the overflow of provinces in other blocks, but also sends out relations to external provinces and acts as a "bridge" intermediary in the overall network. For example, Hunan is a bridge zone connecting the eastern coastal identity with the western inland provinces. Chongqing is an important channel for southwest China to connect east and west and connect north and south.

In order to further explore the spatial correlation of electronic information manufacturing industry among the four blocks, this paper uses the distribution results in Table 2 to calculate the network density matrix of each block. In addition, according to the previous calculation, the spatial correlation density of the national electronic information manufacturing industry in 2016 is 0.2586. If the network density of any one of the four blocks is greater than 0.2586, it means that the development of electronic information manufacturing industry is more concentrated in this block and has a value of 1, and conversely, it has a value of 0, thus forming the image matrix of the spatial correlation network in the electronic information manufacturing industry. The density matrix and image matrix are shown in Table 3. This paper analyzes the correlation among the four major blocks: The first block, the third block and the fourth block have their own internal relations of electronic information manufacturing industry. In addition, the first block accepts the overflow of the second block; The third block also accepts spillovers from the remaining three blocks, which indicates that the more the regional economy develops, the more developed the electronic information manufacturing industry is, and the more technical resources from other provinces are needed.

**Table 3.** Spatial correlation block density matrix and image matrix of the inter-provincial electronic information manufacturing industry.

| Block | Density Matrix | | | | Image Matrix | | | |
|---|---|---|---|---|---|---|---|---|
| | Block 1 | Block 2 | Block 3 | Block 4 | Block 1 | Block 2 | Block 3 | Block 4 |
| Block 1 | 0.714 | 0.000 | 0.321 | 0.000 | 1.000 | 0.000 | 1.000 | 0.000 |
| Block 2 | 0.429 | 0.250 | 0.531 | 0.196 | 1.000 | 0.000 | 1.000 | 0.000 |
| Block 3 | 0.036 | 0.000 | 0.768 | 0.107 | 0.000 | 0.000 | 1.000 | 0.000 |
| Block 4 | 0.000 | 0.000 | 0.429 | 0.452 | 0.000 | 0.000 | 1.000 | 1.000 |

## 4. Conclusions and Policy Recommendations

Based on the statistical data of inter-provincial electronic information manufacturing industry from 2007 to 2016, this paper re-examines the spatial correlation of regional development of electronic information manufacturing industry from a network perspective and conducts an empirical study on the spatial correlation network structure of inter-provincial electronic information manufacturing industry by constructing a modified gravity model and using social network analysis method. The main findings are as follows:

(1) Considering the development of electronic information manufacturing industry in a large area is a necessary condition for the sustainable development of industrial regional economy. Thus, from the perspective of the overall network structure characteristics, the inter-provincial electronic information manufacturing industry in China presents a typical network structure. During the sample period, the spatial network correlation degree showed an upward trend but the density value was low. The overall accessibility of the network is strong and there are no isolated provinces. The network structure is stable, the mutual influence among regions is gradually strengthened, and the regional barriers are gradually broken down. It is conducive to the promotion of inter-provincial technical exchanges and cooperation in electronic and information manufacturing, the development of regional industrial agglomeration benefits. It is conducive to the steady development of regional industrial structure in the direction of rationalization and upgrading.

(2) Mastering the position and role of each province in the spatial association network of electronic information manufacturing industry is conducive to the targeted implementation of regional industrial policies, thus promoting coordinated and sustainable development of regional economy. Therefore, from the analysis of network individual characteristics and block model, eastern provinces with developed economy and good resource endowments, such as Jiangsu and Guangdong, rank high in the measurement results of centrality, are in the central position in the spatial association network of inter-provincial electronic information manufacturing industry. They have close spatial association with other provinces and have strong influence. They can effectively attract relevant resources and elements and obtain more spillover relationships from other provinces. Provinces with relatively weak economic development level, such as Liaoning and Qinghai, are all ranked lower in the centrality measurement results and are located at the periphery of the network structure. Their ability to control and influence other units needs to be strengthened, with obvious overflow. Provinces with unique geographical location, such as Hunan and Chongqing, have played a "bridge" role in the network, promoting the flow of development momentum of electronic information manufacturing industry among regions. Beijing, Tianjin and other provinces have benefited from the development of other provinces, while exporting resources to other provinces. The two-way effect is significant. The resource diffusion and spillover effects of China's inter-provincial electronic information manufacturing industry are conducive to giving full play to the radiation effect of various regions, promoting the complementary industrial advantages of neighboring provinces and cities, sharing innovative technologies and the flow of high-tech talents. It helps to promote the optimal allocation of resources, ensure fairness and coordination among regions, and achieve sustainable economic development.

According to the above conclusions, this paper puts forward the following policy recommendations:

First, on the whole, local governments should establish a global concept and systematic thinking, conform to the evolution trend of the spatial pattern of the development of electronic information manufacturing industry among provinces, and regard spatial correlation and characteristics as important reference factors for their formulation of regional coordinated development policies of electronic information manufacturing industry. In view of the low spatial correlation density of the inter-provincial electronic information manufacturing industry, each province and city should, according to their respective resource endowment and industrial development level, appropriately enhance the inter-provincial industrial interaction and cooperation and improve the spatial correlation degree on the premise of avoiding the phenomenon of diseconomy of scale. Provinces and cities located at the center of the spatial correlation network of electronic information manufacturing industry should actively exert their own radiation-driven effects to promote complementary industrial advantages, sharing of innovative technologies, and the flow of high-tech talents in neighboring provinces and cities. They should break the bad situation of fragmented government, market segmentation, and local protectionism [43]. In the process of continuously optimizing the spatial connection network of China's electronic information manufacturing industry, we should make full use of the "information bridge" function of social relations to form the cohesion of the entire regional economic subject, promote new industrial agglomeration, and realize the stable and sustainable development of the regional economy.

Secondly, from the perspective of nodes and blocks, the spatial layout of China's electronic information manufacturing industry should be optimized by making use of differentiated industrial policies according to the different status of provinces in the spatial correlation network and the transmission mechanism inside and outside the blocks. For the provinces located in the "bidirectional spillover block" and "net beneficial block", the industrial structure optimization and adjustment should be strengthened to promote the reverse effect of the beneficiary provinces to the spillover provinces. Among them, the electronic and information manufacturing industries in Beijing and Shanghai have shown a relative decline, but the information service industries in the two places are very developed and are an important window for China's economic development abroad. Therefore, the focus of the industrial policies of the two places is to speed up the development of software and information service industry, which can make full use of the existing development foundation of the electronic information manufacturing industry of the two places and provide supporting to the surrounding areas. Jiangsu and Guangdong have been in the leading position in China's electronic information manufacturing industry for many years, and their share of output value ranks in the top two in the country. However, the high degree of agglomeration in the two places is not conducive to the continuous growth of the industry. Therefore, they should take the initiative to speed up the transfer of low-end manufacturing links to other provinces, continuously enhance their technological innovation capability, and lead the industry to develop in a high-end direction. For provinces located in the "net spillover block" and "broker block", they should give full play to their advantages in resources, environmental ecology, and other aspects, actively carry out horizontal and vertical cooperation with industrial developed regions, realize cross-regional electronic information manufacturing industry consolidation, and improve the efficiency of the national spatial correlation network. Among them, Hubei, Sichuan, Chongqing and other emerging areas of electronic information industry have strong supporting and cooperative capabilities and have gradually built a complete industrial chain including IC (Integrated Circuit) design, chip manufacturing, packaging and testing, and materials and equipment. These areas should take advantage of their own advantages to undertake industrial transfer in the east and improve their position in the industrial chain. In addition, for provinces such as Inner Mongolia and Qinghai, where the development of electronic information manufacturing industry is relatively slow, they do not yet have the advantages to undertake industrial transfer. They should combine the local resource to speed up the cultivation of local characteristic industries. In short, different provinces and different blocks need to play their respective functions to achieve coordinated and sustainable development of regional economy.

Regional sustainable development requires comprehensive consideration of economic, environmental and social factors in order to pursue sustainable, healthy and harmonious development. The world electronic and information manufacturing industry has entered a new era of rapid development. "Made in China 2025" and "Labor 4.0" are both important strategic measures for the development of manufacturing industry under the background of the new round of scientific and technological revolution and industrial transformation. Among them, Germany's implementation of the "Industrial 4.0" strategy has well grasped the trend of a new round of industrial revolution. It not only gives full play to its advantages in manufacturing, but also actively embraces the integration and innovation of a new generation of emerging green manufacturing technologies, which has important reference significance for China to take the new road of industrialization with Chinese characteristics. Therefore, we should take advantage of the new opportunity of China's "Made in China 2025" to implement the manufacturing power strategy, adhere to the basic policy of "innovation-driven, quality first, green development, structural optimization, and talent-oriented", and promote the development of electronic information manufacturing industry in the direction of intelligence, digitization, Internet of Things, and greening. Specifically, China should support research and development and innovation of core key technologies in the industry, promote demonstration and application of key products, enhance systematic innovation capability, and lay a solid foundation for industrial development, so as to ensure that the development of electronic information manufacturing industry matches the international technological development situation. China should also pay attention to research and development of substitute environmental protection materials to produce electronic products, adopt green equipment and technological processes, change the traditional development mode, promote "green manufacturing". We should persist in taking sustainable development as an important focus for building a manufacturing power, follow the development path of ecological civilization, and realize sustainable development of regional environment. Besides, high-quality education is the ultimate guarantee for scientific and technological innovation and scientific and technological products, and is the fundamental driving force for the sustainable development of labor factors. Therefore, it is necessary to increase the training of high-end management and professional and technical personnel in the electronic information manufacturing industry and make a good reserve of talents for industrial development. In addition, we should improve the electronic information market, strengthen the construction of electronic industrial parks, bring more sufficient policies, funds, infrastructure, talents and other support for the electronic information manufacturing industry, so as to speed up the construction of social informatization, drive social development, and finally realize comprehensive and sustainable development based on environmental sustainability, with economic sustainability as the prerequisite and social sustainability as the goal.

**Author Contributions:** The study was designed by Y.Z. The data were collected by Z.L. The results were analyzed by Z.L. and Y.Z. The policies related to the research were reviewed by Z.L. and Y.Z. The writing work of corresponding parts and the major revisions of this paper were completed by Y.Z.

**Funding:** This research was funded by the National Nature Science Foundation of China, grant number 71401019 and Foundation of Academic Leader Training in Sichuan Province, grant number [2015]100-6, and Foundation of Sichuan University, grant number skyb201709. The APC was funded by Foundation of Sichuan University.

**Acknowledgments:** Authors would like to thank the reviewers of this journal for helpful comments and suggestions. This research is funded by the National Nature Science Foundation of China (71401019), Foundation of Academic Leader Training in Sichuan Province ([2015]100-6), Foundation of Sichuan University (skyb201709). Any errors and all views expressed remain our own.

**Conflicts of Interest:** The authors declare no conflict of interest.

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
