# Peer review of "Research on Spatial Correlation Network Structure of Inter-Provincial Electronic Information Manufacturing Industry in China"

_sustainability, doi:10.3390/su11133534_

Round 1

Reviewer 1 Report

Dear authors, surely your research is based on solid foundations and is also well exposed. However, there is no connection with sustainability issues. This is proven by the fact that the term "sustainability" never appears in the text, Only once in line 414 appears the term sustainable development. Undoubtedly, your work can be successfully published in another process management journal.

Author Response

Dear reviewer,

Thank you for your comments concerning our manuscript. Those comments are all valuable and very helpful for revising and improving our paper, as well as the important guiding significance to our research. We have studied comments carefully and have made correction which we hope meet with approval.

We have made changes for the manuscript, as shown in the updated version, and the following are reply to the reviewer point to point.

Point 1: Dear authors, surely your research is based on solid foundations and is also well exposed.

Response 1: Thanks for your comment, your affirmation is my greatest honor.

Point 2However, there is no connection with sustainability issues. This is proven by the fact that the term "sustainability" never appears in the text, only once in line 414 appears the term sustainable development.

Response 2:

We feel very sorry that we did not clearly express our views on sustainability in the paper, so that there was only one "sustainability" in it. In the meantime, we feel very grateful and cherish the opportunity you have given us to revise our paper.

Now we have carefully revised this paper. Since it is a major revision, almost all the red fonts in the updated version have been revised around this comment. The main corrections in the document are marked in red and italics as follows:

l  In the introduction, we restated the relationship between the research on spatial correlation structure of China's inter-provincial electronic information manufacturing industry and sustainable development. The main revised part is as follows (line 44-75 and 126-140 under updated manuscript):

“However, from the current development point of view, due to the differences in economic development level, resource endowment and geographical location of various provinces, the regional development of China's electronic information manufacturing industry is quite different. In 2016, the main business income of the electronic information manufacturing industry in the eastern region accounted for 74.7% of the national total, and the scale of research and development personnel accounted for 70.5% of the national total. On the one hand, unbalanced regional development and widening regional disparities are not conducive to the rational allocation of resources, the rational distribution of industries and the complementary advantages of various regions, and the long-term sustainable and stable development of the national economy. The goal of sustainable and coordinated development is to gradually narrow the regional disparities on the basis of steadily improving the overall benefits of the national economy and finally achieve balanced development [6]. On the other hand, unbalanced regional development does not mean that regions are divided and do not affect each other [7]. In fact, the spatial correlation of the development of China's inter-provincial electronic information manufacturing industry has shown a systematic and complex network structure. From the perspective of government policies, the country attaches great importance to regional coordinated development and has formulated corresponding strategies for regional coordinated development of electronic information manufacturing industry, so as to give full play to regional advantages and promote the correlation between the development of electronic information manufacturing industry in various provinces. From the perspective of market mechanism, the capital, technology, talents and other factors of production in various provinces have realized cross-regional resource flow, which has greatly enhanced the spatial correlation of the development of inter-provincial electronic information manufacturing industry. The purpose of reasonable resource allocation is to achieve social equity, stable ecological environment, balanced social economy and sustainable development [8]. Therefore, studying the spatial correlation structure of inter-provincial electronic information manufacturing industry can not only master the evolution trend of the overall network spatial pattern, but also clarify the position and role of each province in the spatial correlation network of electronic information manufacturing industry and the spillover effect of network space inside and outside the region, thus implementing regional industrial policies in a targeted way, promoting inter-provincial industrial agglomeration and technology spillover of electronic information manufacturing industry, improving the optimal allocation level of national resources, and finally realizing the sustainable development of regional economy.”

“As a new research paradigm [22], social network analysis has been widely used in sociology [23, 24], economics [25, 26], management [27, 28]. In addition, some studies use this method to explore regional spatial network research [29, 30].Based on relational data and network perspective, this paper constructs a spatial correlation network model using inter-provincial electronic information manufacturing data from 2007 to 2016, and explores the spatial correlation of China's electronic information manufacturing industry using social network analysis. By measuring network density, network connectedness, network hierarchy and network efficiency, the overall network characteristics and evolution trend are analyzed [31]. Through measuring degree centrality and betweenness centrality, this paper analyzes the position and role of individual provinces in the network and masters the internal connections and differences between regions [32]. Through spatial clustering, the provinces are divided into four blocks, the attribute characteristics of each block and the transmission mechanism of spillover effects within and between blocks are analyzed, and then policy suggestions are put forward to promote the coordinated development and overall promotion of regional industries, so as to provide theoretical guidance and decision-making reference for the sustainable development of China's electronic information manufacturing industry.”

l  In the conclusion part, we focus on the future impacts related to sustainability. The revised conclusions are as follows (line 416-448 under updated manuscript):

    “(1) Considering the development of electronic information manufacturing industry in a large area is a necessary condition for the sustainable development of the industry, so from the perspective of the overall network structure characteristics, the inter-provincial electronic information manufacturing industry in China presents a typical network structure. During the sample period, the spatial network correlation degree showed an upward trend but the density value was low. The overall accessibility of the network is strong and there are no isolated provinces. The network structure is stable, the mutual influence among regions is gradually strengthened, and the regional barriers are gradually broken down. It is conducive to the promotion of inter-provincial technical exchanges and cooperation in electronic and information manufacturing, the development of regional industrial agglomeration benefits, and the steady development of regional industrial structure in the direction of rationalization and upgrading.

(2) Mastering the position and role of each province in the spatial association network of electronic information manufacturing industry is conducive to the targeted implementation of regional industrial policies, thus promoting regional coordinated and sustainable development.  Therefore, from the analysis of network individual characteristics and block model, eastern provinces with developed economy and good resource endowments, such as Jiangsu and Guangdong, rank high in the measurement results of centrality, are in the central position in the spatial association network of inter-provincial electronic information manufacturing industry, have close spatial association with other provinces, have strong influence, can effectively attract relevant resources and elements, and obtain more spillover relationships from other provinces.  Provinces with relatively weak economic development level, such as Liaoning and Qinghai, are all ranked lower in the centrality measurement results and are located at the periphery of the network structure. Their ability to control and influence other units needs to be strengthened, with obvious overflow.  Provinces with unique geographical location, such as Hunan and Chongqing, have played a "bridge" role in the network, promoting the flow of development momentum of electronic information manufacturing industry among regions.  Beijing, Tianjin and other provinces have benefited from the development of other provinces, while exporting resources to other provinces. The two-way effect is significant. The resource diffusion and spillover effects of China's inter-provincial electronic information manufacturing industry are conducive to giving full play to the radiation effect of various regions, promoting the complementary industrial advantages of neighboring provinces and cities, sharing innovative technologies and the flow of high-tech talents, promoting the optimal allocation of resources, ensuring fairness and coordination among regions, and achieving sustainable economic development.”

l  3. In the recommendation part, we have put forward recommendations in a more targeted way and stressed that the ultimate goal is to achieve economic sustainability. The revised policy recommendations are as follows (line 451-527 under updated manuscript):

“First, on the whole, local governments should establish a global concept and systematic thinking, conform to the evolution trend of the spatial pattern of the development of electronic information manufacturing industry among provinces, and regard spatial correlation and characteristics as important reference factors for their formulation of regional coordinated development policies of electronic information manufacturing industry. In view of the low spatial correlation density of the inter-provincial electronic information manufacturing industry, each province and city should, according to their respective resource endowment and industrial development level, appropriately enhance the inter-provincial industrial interaction and cooperation and improve the spatial correlation degree on the premise of avoiding the phenomenon of diseconomy of scale. Provinces and cities located at the center of the spatial correlation network of electronic information manufacturing industry should actively exert their own radiation-driven effects to promote complementary industrial advantages, sharing of innovative technologies and the flow of high-tech talents in neighboring provinces and cities, and break the bad situation of fragmented government, market segmentation and local protectionism[43]. In the process of continuously optimizing the spatial correlation network of China's electronic information manufacturing industry, the optimal allocation of resources will be realized to promote the sustainable and coordinated development of electronic information manufacturing industry among provinces.

Secondly, from the perspective of nodes and blocks, the spatial layout of China's electronic information manufacturing industry should be optimized by making use of differentiated industrial policies according to the different status of provinces in the spatial correlation network and the transmission mechanism inside and outside the blocks. For the provinces located in the “bidirectional spillover block” and “net beneficial block”, the industrial structure optimization and adjustment should be strengthened to promote the reverse effect of the beneficiary provinces to the spillover provinces.  Among them, the electronic and information manufacturing industries in Beijing and Shanghai have shown a relative decline, but the information service industries in the two places are very developed and are an important window for China's economic development abroad. Therefore, the focus of the industrial policies of the two places is to speed up the development of software and information service industry, which can make full use of the existing development foundation of the electronic information manufacturing industry of the two places and provide supporting support to the surrounding areas. Jiangsu and Guangdong have been in the leading position in China's electronic information manufacturing industry for many years, and their share of output value ranks in the top two in the country. However, the high degree of agglomeration in the two places is not conducive to the continuous growth of the industry. Therefore, they should take the initiative to speed up the transfer of low-end manufacturing links to other provinces, continuously enhance their technological innovation capability, and lead the industry to develop in a high-end direction. For provinces located in the “net spillover block” and “broker block”, they should give full play to their advantages in resources, environmental ecology and other aspects, actively carry out horizontal and vertical cooperation with industrial developed regions, realize cross-regional electronic information manufacturing industry consolidation, and improve the efficiency of the national spatial correlation network. Among them, Hubei, Sichuan, Chongqing and other emerging areas of electronic information industry have strong supporting and cooperative capabilities, and have gradually built a complete industrial chain including IC design, chip manufacturing, packaging and testing, and materials and equipment. These areas should take advantage of their own advantages to undertake industrial transfer in the east and improve their position in the industrial chain. In addition, for provinces such as Inner Mongolia and Qinghai, where the development of electronic information manufacturing industry is relatively slow, they do not yet have the advantages to undertake industrial transfer. They should combine the local resource to speed up the cultivation of local characteristic industries. In short, different provinces and different blocks need to play their respective functions to achieve coordinated and sustainable development of regional economy.

Finally, we should take advantage of the new opportunity of China's "Made in China 2025" to implement the manufacturing power strategy, adhere to the basic policy of "innovation-driven, quality first, green development, structural optimization, and talent-oriented", and promote the development of electronic information manufacturing industry in the direction of intelligence, digitization, Internet of Things, and greening. The world electronic information industry has entered a new era of rapid development and is increasingly driven by research and development. Germany has well grasped the trend of the new round of industrial revolution. It not only gives full play to its advantages in the manufacturing industry, but also actively embraces the integration and innovation of new generation of emerging technologieswhich has important reference significance for China to accelerate the transformation and upgrading of electronic information manufacturing industry by using "Made in China 2025". China should support research and development and innovation of core key technologies in the industry, promote demonstration and application of key products, enhance systematic innovation capability, and lay a solid foundation for industrial development, so as to ensure that the development of electronic information manufacturing industry matches the international technological development situation, and pay attention to research and development of substitute environmental protection materials to produce electronic products to realize "green manufacturing". Besides, high-quality education is the ultimate guarantee for scientific and technological innovation and scientific and technological products, and is the fundamental driving force for the sustainable development of labor factors. Therefore, it is necessary to increase the training of high-end management and professional and technical personnel in the electronic information manufacturing industry, and make a good reserve of talents for industrial development. In addition, we should improve the electronic information market, strengthen the construction of electronic industrial parks, bring more sufficient policies, funds, infrastructure, talents and other support for the electronic information manufacturing industry, and provide strong support for accelerating the construction of social informatization and the sustainable development of the national economy.”

We tried our best to improve the manuscript and made some changes in the manuscript.

Once again, we appreciate for reviewers’ warm work earnestly, and hope that the revised version will meet with approval.

Yangjingjing Zhang

Sichuan University

2019.6.13

Reviewer 2 Report

The study of spatial correlation of regional development of electronic information manufacturing industry is an interesting topic from the scientific and industrial point of view; however, unfortunately, the research developed by the authors presents certain limitations.

The paper presents a scientific methodology based on the application of a modified gravity model complemented with a social network analysis method taking as data the statistical values for the period (2009-2016). However, the authors unfortunately present a set of conclusions of a general nature for the industrial companies, especially in relation to the final recommendations. It is recommended to the authors to focus on detailing the conclusions of the presented research so that the paper shows accurately the future impact related to sustainability  in case that the industrial companies do not follow the presented recomendations.

Author Response

Dear reviewer,

Thank you for your comments concerning our manuscript. Those comments are all valuable and very helpful for revising and improving our paper, as well as the important guiding significance to our research. We have studied comments carefully and have made correction which we hope meet with approval.

We have made changes for the manuscript, as shown in the updated version, and the following are reply to the reviewer point to point.

Point 1: The study of spatial correlation of regional development of electronic information manufacturing industry is an interesting topic from the scientific and industrial point of view.

Response 1: Thanks for your comment, your affirmation is my greatest honor.

Point 2The research developed by the authors presents certain limitations. The authors unfortunately present a set of conclusions of a general nature for the industrial companies, especially in relation to the final recommendations. It is recommended to the authors to focus on detailing the conclusions of the presented research so that the paper shows accurately the future impact related to sustainability in case that the industrial companies do not follow the presented recomendations.

Response 2: Thank you for your valuable comment, the comment you provided is very valuable for us. We feel very sorry that our conclusion is too general.  According to your suggestion, we have carefully reorganized and improved part 4(Conclusions and Policy Recommendations) of the paper. 

In the conclusion part, we focus on the future impacts related to sustainability. The revised conclusions are as follows (line 416-448 under updated manuscript):

    “(1) Considering the development of electronic information manufacturing industry in a large area is a necessary condition for the sustainable development of the industry, so from the perspective of the overall network structure characteristics, the inter-provincial electronic information manufacturing industry in China presents a typical network structure. During the sample period, the spatial network correlation degree showed an upward trend but the density value was low. The overall accessibility of the network is strong and there are no isolated provinces. The network structure is stable, the mutual influence among regions is gradually strengthened, and the regional barriers are gradually broken down. It is conducive to the promotion of inter-provincial technical exchanges and cooperation in electronic and information manufacturing, the development of regional industrial agglomeration benefits, and the steady development of regional industrial structure in the direction of rationalization and upgrading.

(2) Mastering the position and role of each province in the spatial association network of electronic information manufacturing industry is conducive to the targeted implementation of regional industrial policies, thus promoting regional coordinated and sustainable development.  Therefore, from the analysis of network individual characteristics and block model, eastern provinces with developed economy and good resource endowments, such as Jiangsu and Guangdong, rank high in the measurement results of centrality, are in the central position in the spatial association network of inter-provincial electronic information manufacturing industry, have close spatial association with other provinces, have strong influence, can effectively attract relevant resources and elements, and obtain more spillover relationships from other provinces.  Provinces with relatively weak economic development level, such as Liaoning and Qinghai, are all ranked lower in the centrality measurement results and are located at the periphery of the network structure. Their ability to control and influence other units needs to be strengthened, with obvious overflow.  Provinces with unique geographical location, such as Hunan and Chongqing, have played a "bridge" role in the network, promoting the flow of development momentum of electronic information manufacturing industry among regions.  Beijing, Tianjin and other provinces have benefited from the development of other provinces, while exporting resources to other provinces. The two-way effect is significant. The resource diffusion and spillover effects of China's inter-provincial electronic information manufacturing industry are conducive to giving full play to the radiation effect of various regions, promoting the complementary industrial advantages of neighboring provinces and cities, sharing innovative technologies and the flow of high-tech talents, promoting the optimal allocation of resources, ensuring fairness and coordination among regions, and achieving sustainable economic development.”

In the recommendation part, we have mainly revised the differences and resources supporting parts and put forward countermeasures in a more targeted way. The revised policy recommendations are as follows (line 469-527 under updated manuscript):

    “Secondly, from the perspective of nodes and blocks, the spatial layout of China's electronic information manufacturing industry should be optimized by making use of differentiated industrial policies according to the different status of provinces in the spatial correlation network and the transmission mechanism inside and outside the blocks. For the provinces located in the “bidirectional spillover block” and “net beneficial block”, the industrial structure optimization and adjustment should be strengthened to promote the reverse effect of the beneficiary provinces to the spillover provinces.  Among them, the electronic and information manufacturing industries in Beijing and Shanghai have shown a relative decline, but the information service industries in the two places are very developed and are an important window for China's economic development abroad. Therefore, the focus of the industrial policies of the two places is to speed up the development of software and information service industry, which can make full use of the existing development foundation of the electronic information manufacturing industry of the two places and provide supporting support to the surrounding areas. Jiangsu and Guangdong have been in the leading position in China's electronic information manufacturing industry for many years, and their share of output value ranks in the top two in the country. However, the high degree of agglomeration in the two places is not conducive to the continuous growth of the industry. Therefore, they should take the initiative to speed up the transfer of low-end manufacturing links to other provinces, continuously enhance their technological innovation capability, and lead the industry to develop in a high-end direction. For provinces located in the “net spillover block” and “broker block”, they should give full play to their advantages in resources, environmental ecology and other aspects, actively carry out horizontal and vertical cooperation with industrial developed regions, realize cross-regional electronic information manufacturing industry consolidation, and improve the efficiency of the national spatial correlation network. Among them, Hubei, Sichuan, Chongqing and other emerging areas of electronic information industry have strong supporting and cooperative capabilities, and have gradually built a complete industrial chain including IC design, chip manufacturing, packaging and testing, and materials and equipment. These areas should take advantage of their own advantages to undertake industrial transfer in the east and improve their position in the industrial chain. In addition, for provinces such as Inner Mongolia and Qinghai, where the development of electronic information manufacturing industry is relatively slow, they do not yet have the advantages to undertake industrial transfer. They should combine the local resource to speed up the cultivation of local characteristic industries. In short, different provinces and different blocks need to play their respective functions to achieve coordinated and sustainable development of regional economy.

Finally, we should take advantage of the new opportunity of China's "Made in China 2025" to implement the manufacturing power strategy, adhere to the basic policy of "innovation-driven, quality first, green development, structural optimization, and talent-oriented", and promote the development of electronic information manufacturing industry in the direction of intelligence, digitization, Internet of Things, and greening. The world electronic information industry has entered a new era of rapid development and is increasingly driven by research and development. Germany has well grasped the trend of the new round of industrial revolution. It not only gives full play to its advantages in the manufacturing industry, but also actively embraces the integration and innovation of new generation of emerging technologieswhich has important reference significance for China to accelerate the transformation and upgrading of electronic information manufacturing industry by using "Made in China 2025". China should support research and development and innovation of core key technologies in the industry, promote demonstration and application of key products, enhance systematic innovation capability, and lay a solid foundation for industrial development, so as to ensure that the development of electronic information manufacturing industry matches the international technological development situation, and pay attention to research and development of substitute environmental protection materials to produce electronic products to realize "green manufacturing". Besides, high-quality education is the ultimate guarantee for scientific and technological innovation and scientific and technological products, and is the fundamental driving force for the sustainable development of labor factors. Therefore, it is necessary to increase the training of high-end management and professional and technical personnel in the electronic information manufacturing industry, and make a good reserve of talents for industrial development. In addition, we should improve the electronic information market, strengthen the construction of electronic industrial parks, bring more sufficient policies, funds, infrastructure, talents and other support for the electronic information manufacturing industry, and provide strong support for accelerating the construction of social informatization and the sustainable development of the national economy.”

We tried our best to improve the manuscript and made some changes in the manuscript.

Once again, we appreciate for reviewers’ warm work earnestly, and hope that the revised version will meet with approval.

Yangjingjing Zhang

Sichuan University

2019.6.13

Reviewer 3 Report

- The introduction is rather low on references and could be extended accordingly, adding further works that support its argumentation.This literature should then also be discussed later on.

- A short characterization of the (most relevant) regions described might make sense for an international readership.

- The results could be placed better within the "Made in China 2025" program (as the authors acknowledge themselves), also including a perspective outside of China by adding, for instance:

Beier, G.; Niehoff, S.; Ziems, T.; Xue, B. Sustainability aspects of a digitalized industry – A comparative study from China and Germany. Int. J. Pr. Eng. Man.-GT. 2017, 4, 227–234, DOI: 10.1007/s40684-017-0028-8.

Müller, J.M.; Voigt, K.-I. Sustainable Industrial Value Creation in SMEs: A Comparison between Industry 4.0 and Made in China 2025. Int. J. Pr. Eng. Man.-GT., 2018, 5, 659-670, 10.1007/s40684-018-0056-z.

Author Response

Dear reviewer,

Thank you for your comments concerning our manuscript. Those comments are all valuable and very helpful for revising and improving our paper, as well as the important guiding significance to our research. We have studied comments carefully and have made correction which we hope meet with approval.

We have made changes for the manuscript, as shown in the updated version, and the following are reply to the reviewer point to point.

Point 1: The introduction is rather low on references and could be extended accordingly, adding further works that support its argumentation. This literature should then also be discussed later on.

Response 1: Thank you for your valuable comment, according to your suggestions, we have improved the Introduction part of the updated version and expanded the literature in the introduction to 32 articles. The added references have been marked in red in the References part of the paper. We also discussed these references later in the paper. For example, the eighth document cited in the introduction (line 66-67 under updated manuscript) also has relevant contents discussed in the Conclusions and Policy Recommendations part (line 443-448 under updated manuscript).

Point 2A short characterization of the (most relevant) regions described might make sense for an international readership.

Response 2: Thank you for your valuable comment, according to your suggestions, we have briefly described the most relevant areas described (line 475-477,481-484 and 491-494 under updated manuscript).

Point 3The results could be placed better within the "Made in China 2025" program (as the authors acknowledge themselves), also including a perspective outside of China by adding, for instance:

Beier, G.; Niehoff, S.; Ziems, T.; Xue, B. Sustainability aspects of a digitalized industry – A comparative study from China and Germany. Int. J. Pr. Eng. Man.-GT. 2017, 4, 227–234, DOI: 10.1007/s40684-017-0028-8.

Müller, J.M.; Voigt, K.-I. Sustainable Industrial Value Creation in SMEs: A Comparison between Industry 4.0 and Made in China 2025. Int. J. Pr. Eng. Man.-GT., 2018, 5, 659-670, 10.1007/s40684-018-0056-z.

Response 3: Thank you for your valuable comment. We have read the above two documents carefully and benefited a lot, and cited them in the paper. We also have included the "Made in China 2025" in the results and added a perspective outside of China in our paper (line 502-518 under updated manuscript).

We tried our best to improve the manuscript and made some changes in the manuscript.

Once again, we appreciate for reviewers’ warm work earnestly, and hope that the revised version will meet with approval.

Yangjingjing Zhang

Sichuan University

2019.6.13

Round 2

Reviewer 1 Report

Dear authors, I appreciate the effort you have made to revise your paper. However, before its publication I recommend two further details:

1. you refer to the concept of sustainability and/or sustainable development in a somewhat general form, which risks becoming abstract. For example, you should specify in which dimension of sustainability the Network Structure affects: the environment, the economy or society? The concept of sustainable development should refer to the simultaneous integration of the three dimensions of sustainability. It should therefore be made clear how this is to be done.  

2. the 4.0 industry paradigm has been shown to be an enabling factor for sustainability, as demonstrated by many literature-based research. On this basis, you could develop the concept of sustainable development related to the "Made in China 2025" plan by linking the two industrial policies and the action they can take on sustainability.

Kind regards

Author Response

Response to Reviewer 1 Comments

Dear reviewer,

Thank you for your comments concerning our manuscript. Those comments are all valuable and very helpful for revising and improving our paper, as well as the important guiding significance to our research. We have studied comments carefully and have made correction which we hope meet with approval.

We have made changes for the manuscript, as shown in the updated version, and the following are reply to the reviewer point to point.

Point 1: You refer to the concept of sustainability and/or sustainable development in a somewhat general form, which risks becoming abstract. For example, you should specify in which dimension of sustainability the Network Structure affects: the environment, the economy or society?

Response 1: Thank you for your valuable comment, the comment you provided is very valuable for us. In our paper, network structure mainly affects the economic dimension of sustainability, and we have also revised the corresponding places in the paper (line 11, 53, 140, 417, 429-430 under updated manuscript).

Point 2The concept of sustainable development should refer to the simultaneous integration of the three dimensions of sustainability. It should therefore be made clear how this is to be done. 

Response 2: Thank you for your valuable comment, according to your suggestions, we have made corresponding modifications in the paper. The revised parts are as follows:

“In the process of continuously optimizing the spatial connection network of China's electronic information manufacturing industry, we should make full use of the "information bridge" function of social relations to form the cohesion of the entire regional economic subject, promote new industrial agglomeration and realize the stable and sustainable development of the regional economy.”  (line 464-468 under updated manuscript)

“Regional sustainable development requires comprehensive consideration of economic, environmental and social factors in order to pursue sustainable, healthy and harmonious development.”  (line 502-504 under updated manuscript)

“China should also pay attention to research and development of substitute environmental protection materials to produce electronic products, adopt green equipment and technological processes, change the traditional development mode, promote "green manufacturing". We should persist in taking sustainable development as an important focus for building a manufacturing power, follow the development path of ecological civilization, and realize sustainable development of regional environment. ”  (line 521-526 under updated manuscript)

“so as to speed up the construction of social informatization, drive social development, and finally realize comprehensive and sustainable development based on environmental sustainability, with economic sustainability as the prerequisite and social sustainability as the goal.”  (line 534-536 under updated manuscript)

Point 3The 4.0 industry paradigm has been shown to be an enabling factor for sustainability, as demonstrated by many literature-based research. On this basis, you could develop the concept of sustainable development related to the "Made in China 2025" plan by linking the two industrial policies and the action they can take on sustainability.

Response 3: Thank you for your valuable comment, according to your suggestions, we have made corresponding modifications in the paper. The revised parts are as follows:

“Regional sustainable development requires comprehensive consideration of economic, environmental and social factors in order to pursue sustainable, healthy and harmonious development.  The world electronic and information manufacturing industry has entered a new era of rapid development. "Made in China 2025" and "Labor 4.0" are both important strategic measures for the development of manufacturing industry under the background of the new round of scientific and technological revolution and industrial transformation. Among them, Germany's implementation of the "Industrial 4.0" strategy has well grasped the trend of a new round of industrial revolution. It not only gives full play to its advantages in manufacturing, but also actively embraces the integration and innovation of a new generation of emerging green manufacturing technologies, which has important reference significance for China to take the new road of industrialization with Chinese characteristics. Therefore, we should take advantage of the new opportunity of China's "Made in China 2025" to implement the manufacturing power strategy, adhere to the basic policy of "innovation-driven, quality first, green development, structural optimization, and talent-oriented", and promote the development of electronic information manufacturing industry in the direction of intelligence, digitization, Internet of Things, and greening.”  (line 502-516 under updated manuscript)

We tried our best to improve the manuscript and made some changes in the manuscript.

Once again, we appreciate for reviewers’ warm work earnestly, and hope that the revised version will meet with approval.

Yangjingjing Zhang

Sichuan University

2019.6.19

Reviewer 2 Report

The authors have addressed satisfactorily the points raised during the review. However, it is recommended that the authors review the wording in English of some sentences that are too long.

Author Response

Response to Reviewer 2 Comments

Dear reviewer,

Thank you for your comments concerning our manuscript. Those comments are all valuable and very helpful for revising and improving our paper, as well as the important guiding significance to our research. We have studied comments carefully and have made correction which we hope meet with approval.

We have made changes for the manuscript, as shown in the updated version, and the following are reply to the reviewer point to point.

Point 1: The authors have addressed satisfactorily the points raised during the review.

Response 1: Thanks for your comment, your affirmation is my greatest honor.

Point 2However, it is recommended that the authors review the wording in English of some sentences that are too long.

Response 2: Thank you for your valuable comment, according to your suggestions, we have made corresponding modifications in the paper (line 52, 72-74, 117, 123, 124, 137, 417, 425, 433, 434, 447,448, 463, 517).

We tried our best to improve the manuscript and made some changes in the manuscript.

Once again, we appreciate for reviewers’ warm work earnestly, and hope that the revised version will meet with approval.

Yangjingjing Zhang

Sichuan University

2019.6.19
